# Senescence and entrenchment in evolution of amino acid sites

A. V. Stolyarova [1✉], E. Nabieva [1,2], V. V. Ptushenko [3,4], A. V. Favorov [5,6], A. V. Popova [7], A. D. Neverov[7] & G. A. Bazykin [1,2]

Amino acid propensities at a site change in the course of protein evolution. This may happen for two reasons. Changes may be triggered by substitutions at epistatically interacting sites elsewhere in the genome. Alternatively, they may arise due to environmental changes that are external to the genome. Here, we design a framework for distinguishing between these alternatives. Using analytical modelling and simulations, we show that they cause opposite dynamics of the fitness of the allele currently occupying the site: it tends to increase with the time since its origin due to epistasis ("entrenchment"), but to decrease due to random environmental fluctuations ("senescence"). By analysing the genomes of vertebrates and insects, we show that the amino acids originating at negatively selected sites experience strong entrenchment. By contrast, the amino acids originating at positively selected sites experience senescence. We propose that senescence of the current allele is a cause of adaptive evolution.

[1] Center of Life Sciences, Skolkovo Institute of Science and Technology, Skolkovo 143028, Russia. [2] Institute for Information Transmission Problems (Kharkevich Institute), Russian Academy of Sciences, Moscow 127051, Russia. [3] Department of Photochemistry and Photobiology, N. M. Emanuel Institute of Biochemical Physics of Russian Academy of Sciences, Moscow 119334, Russia. [4] A. N. Belozersky Institute of Physical–Chemical Biology, M. V. Lomonosov Moscow State University, Moscow 119992, Russia. [5] Division of Biostatistics and Bioinformatics, Department of Oncology, Sidney Kimmel Comprehensive Cancer Center, Johns Hopkins School of Medicine, Baltimore, MD 21205, USA. [6] Laboratory of System Biology and Computational Genetics, Vavilov Institute of General Genetics, Moscow 119991, Russia. [7] Department of Molecular Diagnostics, Central Research Institute for Epidemiology, Moscow 111123, Russia. ✉email: a.stoliarova@skoltech.ru

Fitness landscape is the key concept of evolutionary biology, and its description is necessary to fully understand adaptive evolution and speciation[1–4]. Unfortunately, the large dimensionality of even the landscapes of individual proteins makes them impossible to measure comprehensively in a direct experiment[5,6]. Still, methods of comparative genomics can be used to assess the integral features of fitness landscapes. The simplest informative unit of landscape structure is the single-position fitness landscape (SPFL)[7], i.e., a vector of fitness values of all possible alleles at an individual genomic position. SPFLs change with time[8–15]; this may affect the optimality of the allele that is currently prevalent at this site, influencing subsequent evolution.

One factor entailing changes of SPFL is substitutions at other sites of the genome. For this to be the case, these substitutions need to affect the relative fitness of different variants at the considered site, i.e., these sites have to be involved in epistatic interactions. Epistasis has been postulated to be a prevalent factor of protein evolution and divergence across species[2,6,9,16–24]. One expected manifestation of genome-wide epistasis is entrenchment, or the evolutionary Stokes shift[12,25,26] — a phenomenon whereby the relative fitness of the allele currently prevalent at the site increases as substitutions at interacting sites accumulate. The reason for this increase is the constraint imposed by the site in consideration onto epistatically interacting sites. The evolution of the remaining sequence is constrained to preserve the high fitness of the resident allele, and may even increase it; at the same time, this sequence is free to evolve to become less compatible with other variants not currently present at the site. Over time, this leads to an increase in the fitness of the current allele relative to other alleles, including those that resided at this site earlier.

Entrenchment was demonstrated both in simulated protein evolution[12,25,26] and in evolution of real-life proteins. For example, it was shown that reversals of past substitutions are suggestive of entrenchment: their rate declines with time, indicating that they become more deleterious, i.e., that the current allele becomes more preferable compared to the previous one[27–30]. The decline in the rate of reversals is caused both by the increase in the fitness of the current allele and the decrease in the fitness of the replaced allele[28].

However, the SPFL may change due to environmental fluctuations even in the absence of epistasis. If such changes are recurrent, the fitness landscape becomes a time-dependent "seascape"[16,31–36]. This leads to recurrent positive selection (fluctuating selection) in favor of the newly beneficial alleles and to adaptive evolution[35,37–40]. Nowadays, the way fluctuating selection shapes the dynamics of the relative fitness of the current allele remains poorly studied.

Here, we characterize the effects of epistasis and of fluctuating selection on SPFL changes and estimate the contribution of these forces in past evolution.

## Results

**Environmental fluctuations make the current allele less fit.** First, we ask how fluctuating selection affects the relative fitness of different alleles at a site. If changes of the SPFL are random with regard to the identity of the allele currently residing at the site, we expect that they, on average, will reduce its relative fitness (see formal proof in Supplementary Note 1). This is because, due to natural selection, the relative fitness conferred by the current variant is, on average, higher than that of a random variant at this site. An episode of positive selection triggered by an SPFL change may then cause the spread of a novel variant, which would confer high fitness till the next SPFL change.

To illustrate this, we simulate amino acid evolution on a randomly changing fitness landscape. In this simulation, the fitness values for each of the 20 possible amino acids are drawn from a predefined distribution, and the amino acid substitutions occur with probabilities determined by the corresponding selection coefficients. At random moments of time, fitness values are redrawn from the same distribution (Fig. 1a–c).

As a result of selection, the fitness of the current allele is on average higher than that of other alleles (Fig. 1b, c); in particular, if selection is strong, the site is typically occupied by the best-possible allele (Fig. 1b). However, as the landscape changes randomly, the fitness of this original allele, on average, decreases with time, gradually approaching the mean fitness across all possible variants (Fig. 1e, f). We call this process senescence of the current allele[41]. This effect is more pronounced for the rugged landscape, when one allele is highly more beneficial than others (Fig. 1e), and less pronounced when selection is weaker (Fig. 1f).

The decline in fitness of the current allele due to fluctuating selection leads to an increase in the rate at which it is lost (Fig. 1h, i), in line with the quenched theory of fluctuating selection[34,37] (Supplementary Note 2).

**Senescence speeds up and entrenchment slows down evolution.** Therefore, the two different modes of change of the SPFL are expected to produce the opposite dynamics of the fitness of the allele that currently occupies the site. If the current allele is favored by epistatic interactions with other sites, it will be entrenched, i.e., its fitness, compared to that of other alleles at this site, is expected to increase with time. By contrast, random SPFL changes that occur without regard to the identity of the allele currently occupying the site are expected to decrease its fitness, leading to senescence.

We propose that this dichotomy can be used to distinguish between these two modes of SPFL changes. To infer the changes in the relative fitness of an allele with time, we study the differences in the rate at which it is lost in the course of evolution. Indeed, the relative fitness of an allele specifies the probability that it is substituted by another allele per unit time[42].

Let us assume that a substitution of an ancestral variant A for another variant B (allele gain) has occurred at some internal branch of the phylogenetic tree, and this current allele B has been preserved in several extant species (Fig. 2a). In other species, it could be lost, for example, as a result of a reversal to A or a substitution for some other allele C. If the SPFL for this site has remained static (the fitness of the current allele B has not changed, $\Delta f_B = 0$), the probability of replacement of B per unit time is independent of the time elapsed since its gain.

By contrast, under senescence, the fitness conferred by B decreases with its age ($\Delta f_B < 0$), and the probability of its replacement increases with it. In this case, we will observe a higher rate of substitutions on the branches originating much later than the allele gain, compared to the branches leading to close descendants (Fig. 2a, left). Finally, under entrenchment, the fitness of the current allele increases ($\Delta f_B > 0$), so the rate at which B is lost declines with its age (Fig. 2a, right).

To test the validity of this approach, we used SELVa[43] to simulate molecular evolution at individual sites assuming that the fitness of the allele currently residing at the site changes with time. Specifically, we assume that the log fitness of the current allele is initially drawn from a predefined distribution, and then changes with time linearly with rate $k$. Positive values of $k$ correspond to an increase in the fitness of the current allele, i.e., entrenchment, while negative values correspond to a decrease in its fitness, i.e., senescence (see Methods). As expected, entrenchment ($k > 0$) results in a high rate of substitutions immediately

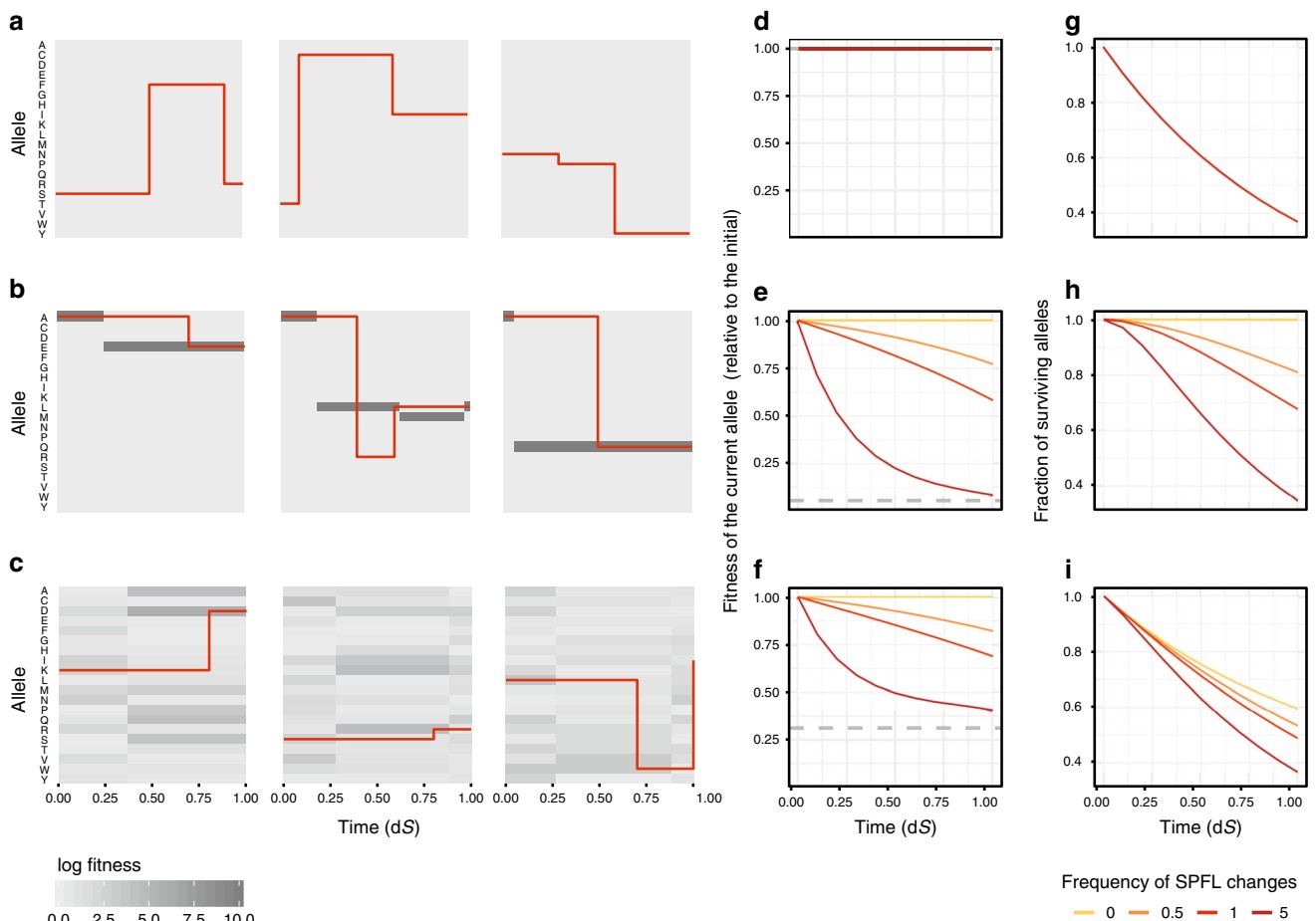

**Fig. 1 Random changes of SPFL reduce the fitness of the current allele. a–c** Examples of how simulated random changes in SPFLs of different shapes provoke allele substitutions. Each of the nine plots shows the history of one simulated amino acid site; red lines represent the current allele at the site, with vertical red lines indicating substitutions. SPFL changes randomly at the average rate of one change per time required for one neutral substitution (1 dS). **a** All 20 possible alleles have the same fitness (flat SPFL), so that all substitutions are neutral. **b** One allele is substantially more beneficial than others (rugged SPFL, log fitness vector = [10, 0, ..., 0]); most of the observed substitutions are positively selected. **c** Log fitness values are drawn from a gamma distribution with shape and rate = 1. **d–f** Changes in the average fitness of the current allele with evolutionary time under random SPFL changes. If the SPFL is static (frequency of SPFL changes = 0), the fitness of the current allele remains constant. The same is true for the flat landscape (**d**). Otherwise, the fitness of the current allele declines for both the rugged (**e**) and the gamma-distributed (**f**) SPFLs, i.e., the current allele undergoes senescence, and the more frequently the SPFL changes, the more rapid is this decline. The mean fitness across all possible alleles is shown with a dashed line. **g–i** The fraction of surviving ancestral alleles as a function of time since the beginning of the simulation. For both the rugged (**h**) and the gamma-distributed (**i**) SPFLs, random changes in the landscape increase the rate at which the original allele is lost (**h**, **i**), unless the SPFL is flat (**g**). For **d–i** 95% confidence bands based on ten repeats are plotted (but too narrow to be seen).

after the ancestral substitution, but a reduced rate later on. By contrast, under senescence ($k < 0$), the rate of substitutions increases with time since the allele gain (Fig. 2b). An alternative mode of simulation of senescence, whereby random changes in SPFL and the molecular evolution caused by them are modeled explicitly, gives the same results (Supplementary Fig. 1).

Besides the phylogenetic distribution of substitutions, the mode and rate of SPFL change also affect the overall rate of molecular evolution (Fig. 2c). Compared to a static landscape of the same shape, entrenchment reduces the substitution rate, as the time-averaged fitness of the current allele is higher, and therefore its substitutions rarer. Conversely, under senescence, many of the substitutions of the current allele are advantageous, increasing the overall rate of evolution. Importantly, senescence doesn't necessarily result in overall evolution rate exceeding the neutral rate, which is a hallmark of positive selection. Indeed, if an allele is strongly preferred, a drop in its fitness over the course of senescence may still leave it the optimal one, so that negative selection will still maintain it (as observed for the rugged SPFL, Fig. 2c right).

**Senescence and entrenchment at single-allele resolution.** Large phylogenies allow detecting changes in substitution frequencies for individual alleles. Each originating allele, e.g., an amino acid arising at a specific site from an ancestral amino acid substitution, can be inherited by multiple descendant lineages leading to different extant species. Ancestral state reconstruction can then be used to infer the lineages at which this allele has been lost due to a reversion or substitution to a different amino acid. If enough such lineages are available, this allows to trace the decline or increase in the rate of allele substitution since its origin, i.e., entrenchment or senescence.

We applied binomial logistic regression to detect changes in substitution frequencies with the age of the current allele along the phylogeny for five mitochondrial genes of Metazoa[14]. The regression was performed separately for each allele B with a known time of origin (corresponding to allele gain A → B) at each site. Among the 42,637 such alleles, we identified 28 alleles for which the frequency of replacement significantly increased with time since their origin (i.e., senescing alleles), and 21 alleles

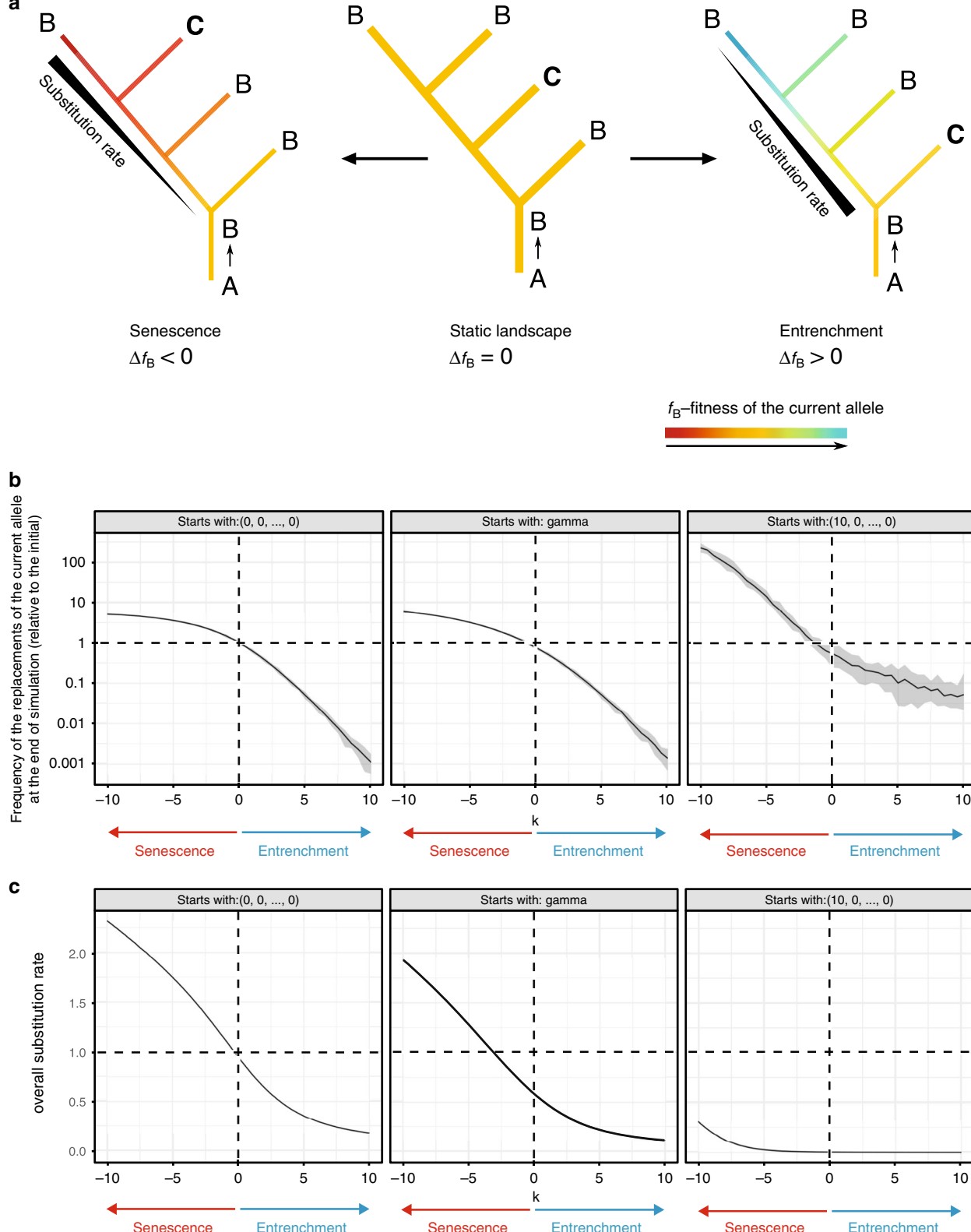

where it decreased (i.e., entrenched alleles) at 5% false discovery rate (Fig. 3a, Supplementary Table 1). The examples of phylogenies indicating allele replacements at senescing and entrenched alleles are shown in Fig. 3b, c. The overall number of substitutions occurring along the phylogeny was similar between sites containing entrenched and senescing alleles (Supplementary Fig. 13, sign test $p$ value = 0.84), indicating that

the substitution rate was insufficient for distinguishing between these scenarios.

**Heterogeneity leads to an artifactual signal of entrenchment.** While phylogenies spanning hundreds and thousands of species, like those available for mitochondrial proteins, allow to measure the

**Fig. 2 Replacement patterns of the current allele reflect changes in its fitness. a** On the static landscape (if the fitness of the current allele B remains constant, $f_B = 0$), the probability that B is replaced per unit time does not depend on the time since its gain A → B. Under entrenchment (right), B becomes more favorable with time ($\Delta f_B > 0$); therefore, the A → B substitution rate declines and there are fewer substitutions observed on "late" branches of the phylogeny. Alternatively, under senescence (left), the fitness of B decreases ($\Delta f_B < 0$), leading to an increase in the rate of its loss with time. **b, c** In simulated evolution, changes in fitness of the current allele affect the dynamics of its replacements (**b**) and the overall substitution rate (**c**). Simulations were started with SPFLs of different shapes: flat SPFL (log fitness vector = [0, 0, …, 0]), rugged SPFL (log fitness vector = [10, 0, …, 0]), and gamma-distributed SPFL. Over the course of simulation, the log fitness of the current allele was linearly changing with time at rate $k$. If there are no changes in the fitness of the current allele ($k = 0$), the frequency of substitutions on the "late" branches is similar to that on the "early" branches (**b**). If the current allele is senescing ($k < 0$), there is an excess of substitutions on the "late" branches (**b**), and the overall substitution rate is increased compared to a static landscape (**c**). Conversely, under entrenchment ($k > 0$), there is a deficit of substitutions on the "late" branches (**b**), and the overall substitution rate is decreased compared to a static landscape (**c**). For **b, c**, mean values and 95% confidence bands based on ten repeats are shown.

changes in the substitution rate for individual alleles, in smaller phylogenies, the number of substitutions experienced by an allele can be insufficient for such an analysis. Still, it may be possible to identify the prevailing patterns of substitutions by pooling alleles together. However, such pooling can be problematic: even in the absence of SPFL changes, the rate of substitution can appear to change with time since allele origin if the pooled alleles have different time-invariant substitution rates, confounding inference of SPFL changes.

Indeed, consider a set of alleles, each characterized by its own substitution rate that is stationary (constant in time) but differs between alleles. While the replacement rate may be constant for each allele, so that the time to replacement is characterized by an exponential distribution, it will not, in general, be exponentially distributed in the resulting heterogeneous dataset. Instead, the frequency of substitution will appear to decline with time (Fig. 4a), making it non-stationary and mimicking entrenchment of the current allele. The problem of data heterogeneity leading to decreasing hazard function is well known in demographic inference[44,45], and has been previously appreciated in the inference of substitution rates dynamics in molecular evolution[28,46]. Notably, no mixture of stationary processes can give rise to an increase in the substitution rate, i.e., senescence[44].

It is obvious that heterogeneity of substitution rates arises from pooling of different amino acid sites. More subtly, it also arises within individual sites as a result of differences between rates of substitution of different alleles. Substitution rate is the product of mutation and fixation rates, and this heterogeneity will arise due to any differences in either of these factors between alleles. For example, consider a single site which is non-neutral, i.e., such that different alleles confer different fitness. Such alleles will be characterized by different replacement rates (lower for high-fitness alleles, and higher for low-fitness alleles), and pooling over different alleles over the course of evolution of this site (or, identically, over different independent and identically distributed sites) would lead to heterogeneity of substitution rates and to an apparent decline in substitution rates with time since allele origin.

To show this, we simulate molecular evolution on static SPFLs of different shapes. If all alleles have the same fitness, i.e., if all substitutions are neutral ("flat" landscape), the substitution rate is independent of time since allele origin (Fig. 4b). By contrast, if fitness values of alleles are drawn from a gamma distribution, so that different alleles have different fitness, the substitution frequency decreases with the age of the current allele, even though the SPFL doesn't change (Fig. 4c). On a more rugged SPFL, when one allele is much more fit than all others, this decline is even sharper (Fig. 4d).

**Inferring senescence and entrenchment from phylogenies.** To address the problem of heterogeneity of alleles, we reconstructed the SPFL dynamics while accounting for differences between alleles in mutation rates and "baseline" selection. In the absence of prior information about the distribution of these

characteristics, it was impossible to reconstruct the explicit likelihood function for the substitution rates. Instead, we used the approximate Bayesian computation (ABC)[47] approach to obtain the posterior distribution of the rate of current allele fitness change per unit time ($k$). ABC depends on a set of summary statistics to evaluate the difference between the simulation results and the data. We used two summary statistics, each aggregating overall individual alleles, which reflect the age-dependent dynamics of substitution rates (see Methods, Supplementary Fig. 2).

We used two models for parameter inference. Under the two-parameter model, we assumed that log fitness values for individual alleles were drawn from a gamma distribution with rate and shape parameters denoted by *alpha*, and the log fitness of the current allele at all sites changed linearly with rate $k$. Under the three-parameter model, the fitness changed linearly only for a fraction of alleles, while the fitness of the remaining alleles was invariant (see Methods for details).

Simulations show that both models perform well in identifying senescence and entrenchment under a broad range of parameters, and are robust to overall substitution rate, phylogeny shape, pooling of sites with diverse characteristics, and errors in ancestral state reconstruction (see Methods, Supplementary Table 2, Supplementary Figs. 3–7).

**Positively selected sites show strong senescence.** We applied the developed ABC approach to protein sequences of vertebrates and insects (Supplementary Fig. 8). To understand how the direction of fitness change depends on the overall conservation of an amino acid site, in both datasets, we classified all codon sites by the type of selection acting at them, on the basis of the ratio of non-synonymous and synonymous substitutions per site ($\omega$): negatively selected ($\omega < 1$), neutral ($\omega = 1$) or positively selected ($\omega > 1$) sites, and analyzed them independently (Table 1). The three-parameter model provided a better fit to the data than the two-parameter model (Supplementary Figs. 9 and 10), so we used the former for the analysis.

In vertebrates, both the fraction of senescing/entrenched alleles and the value of $k$ for them depended on the mode of selection acting at the site. The 36% of alleles originating at negatively selected ($\omega < 1$) sites demonstrated strong evidence for entrenchment: we estimate that all of them are entrenched, indicating that the fitness of the current variant increases with time since its origin (Fig. 5b, left panel; Table 1). By contrast, of the 6% of alleles arising at positively selected sites ($\omega > 1$), 18% experience senescence (Fig. 5b, right panel), indicating a decrease in the fitness of the current allele. While we are unable to distinguish robustly between a low fraction of alleles undergoing strong senescence and a high fraction of alleles undergoing weak senescence, the 95% posterior probability interval does not include $k = 0$, rejecting stationarity. The neutral sites demonstrate an intermediate signal with little evidence for entrenchment or

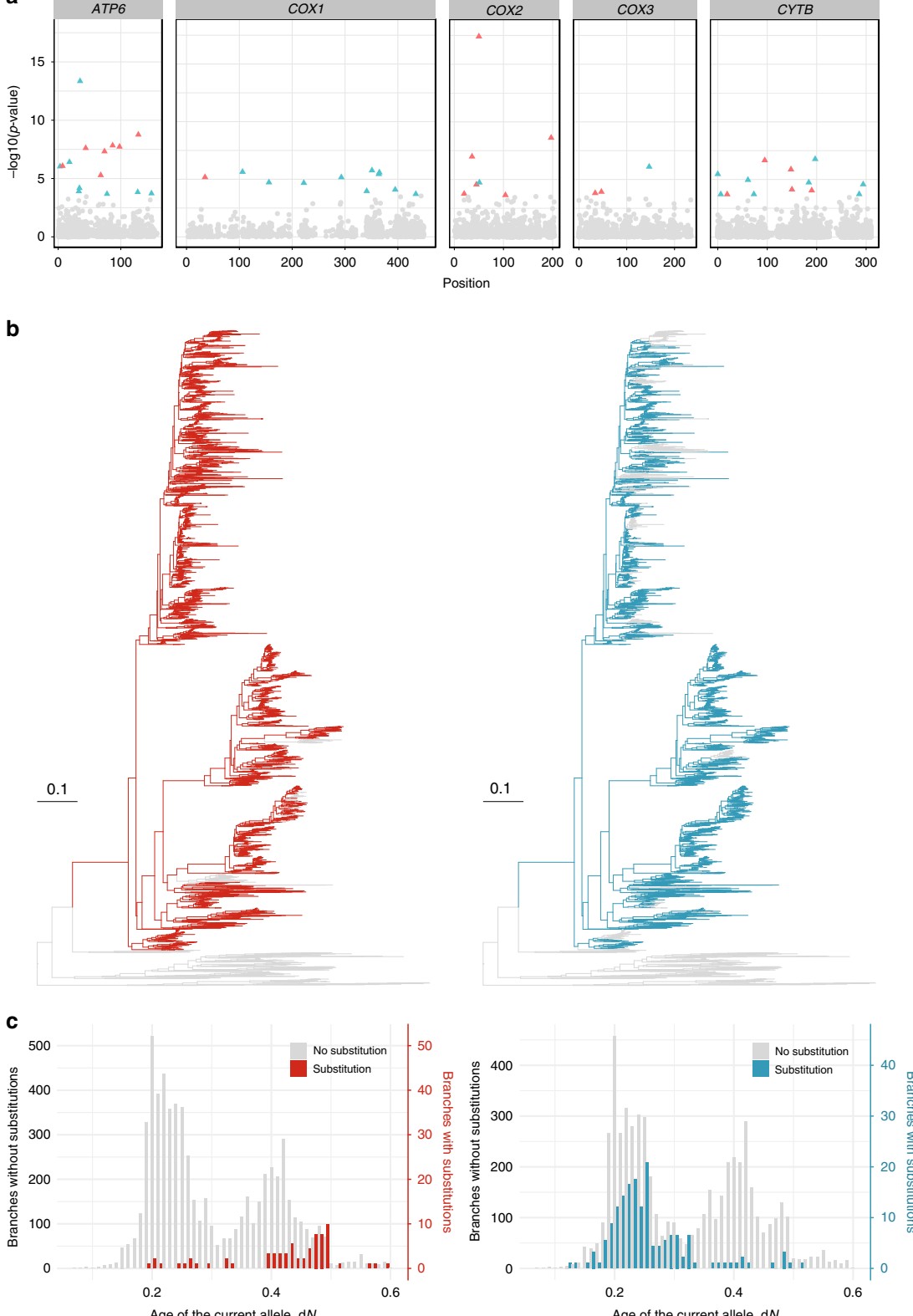

senescence (Fig. 5b, middle panel). A similar pattern is observed in phylogenies of insects (Fig. 5d).

While senescence is observed at sites that undergo rapid substitution ($\omega > 1$), it is distinct from an increase in the overall substitution rate. Similarly, while entrenchment is observed at constrained sites ($\omega < 1$), it is distinct from a reduction in substitution rate. To illustrate this, we simulated evolution under

different substitution rates but constant SPFL on the phylogenies of vertebrates and insects, using the same distribution of $\omega$ values as in the real data, and estimated $k$ using the ABC pipeline (Fig. 5a, c). No senescence or entrenchment was detected for the datasets simulated using the stationary model under $\omega = 1$ or $\omega > 1$. A weak spurious signal of entrenchment was detected for the simulated datasets with $\omega < 1$, resulting from the high

**Fig. 3 Senescence and entrenchment of individual alleles in the mitochondrial genes of Metazoa. a** Manhattan plot of senescing and entrenched alleles. Only the alleles with a known phylogenetic position of origin, i.e., those that were not yet present in the tree root, were analyzed; a single genomic site can contain zero, one or several alleles. *P* values are calculated using binomial logistic regression. The alleles demonstrating significant senescence under 5% FDR are shown in red; the alleles demonstrating entrenchment are shown in green. No amino acid sites contained more than one significantly senescing or entrenched allele. The list of significantly senescing or entrenched alleles is shown in Supplementary Table 1. **b** Examples of senescing (*COX2* position 56, red) and entrenched (*ATP6* position 71, green) alleles. The contiguous segment of the phylogeny carrying the derived allele is shown in color. **c** Distribution of substitutions along the lifetime of alleles shown in (**b**). For the senescing allele, the phylogenetic branches corresponding to allele replacements (red) originate later than the branches without replacements (gray). Conversely, the entrenched allele is more frequently replaced soon after its origin (green).

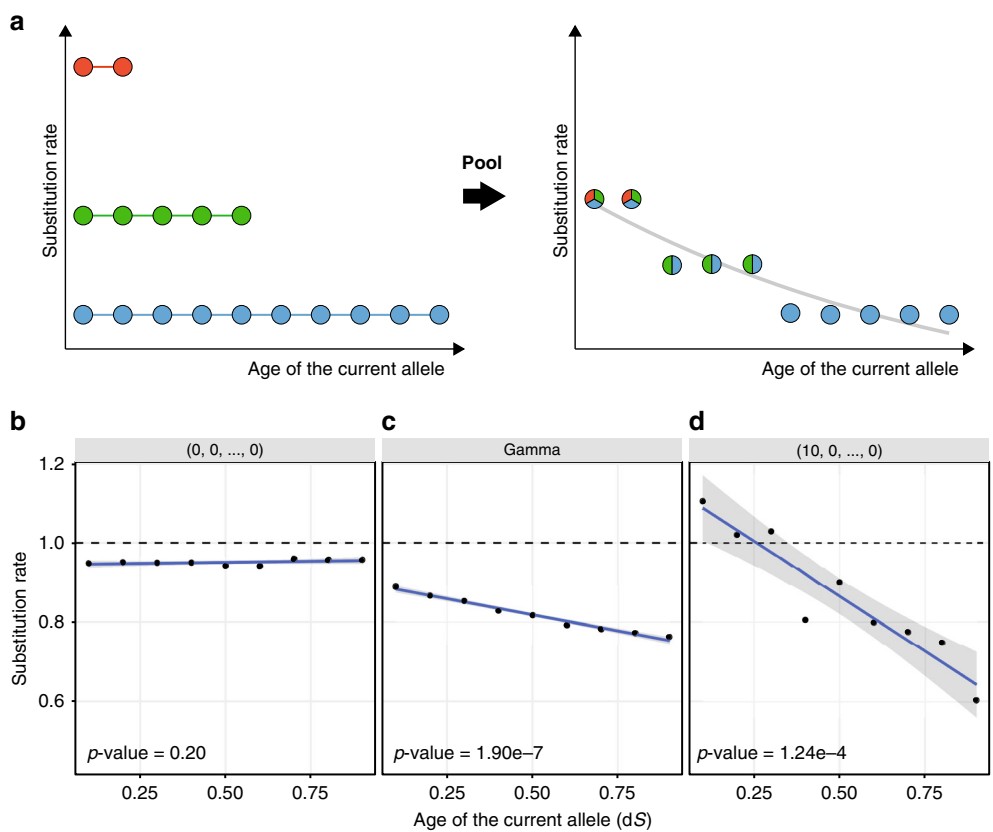

**Fig. 4 Heterogeneity of substitution rates among alleles on static SPFLs imitates entrenchment. a** Consider three classes of alleles, characterized by different constant substitution rates: fast (red), moderate (green), and slow (blue) alleles. For each allele, we can calculate the substitution frequency on the phylogenetic branches located at different evolutionary distances from the gain of that allele (each dot corresponds to a single branch). If the dynamics of replacement of these alleles are analyzed separately for each substitution-rate class, no spurious signal of entrenchment or senescence is observed (left). However, if alleles from different classes are pooled together, the substitution frequency appears to decrease with time, mimicking the signal of entrenchment (right). On a static flat SPFL (**b**, log fitness vector = [0, 0, …, 0]), i.e., when the substitution rates for all 20 alleles are the same, the frequency of replacement of the current allele on some branch of the phylogenetic tree does not depend on the time since the allele gain (each dot corresponds to a single branch). However, on a gamma-distributed SPFL (**c**), the heterogeneity of fitness of the alleles produces entrenchment-like decline of replacement rate of the current allele with time, although the SPFL remains static. The effect is even more pronounced on a more rugged SPFL (**d**, log fitness vector = [10, 0, …, 0]). The *p* values are obtained with linear regression.

heterogeneity in evolution rates (see above); however, it was much weaker than that observed in the data for this category of sites.

## Discussion

While the direction of changes in fitness in the course of evolution is unpredictable for an individual allele, there are certain statistical regularities. Previous work has shown that, at a site involved in epistatic interactions with other sites, the relative fitness conferred by the incumbent allele is expected to increase with time since its origin[9,18,22,23]. Acting alone, i.e., if the overall fitness landscape is static, this process of entrenchment should make the propensities existing at individual sites more pronounced. This, in turn, should limit the level of divergence

between highly divergent sequences, although reaching this level may take a very long time[9,10,48–51].

Here, we consider the dynamics of allele fitness due to changes in the overall fitness landscape itself. We show that, if the direction of these changes is independent of the current position of the population in the genotype space, the expected mean dynamics — senescence — is opposite to that of entrenchment. We design a method to distinguish the two patterns from the phylogenetic distribution of substitutions and find that entrenchment is ubiquitous at negatively selected sites, while senescence is prevalent at sites undergoing adaptive evolution under positive selection. While senescence is a distinct phenomenon from positive selection (Supplementary Note 3), the observed concordance between the direction of non-stationarity

**Table 1 Senescence and entrenchment in evolution of genomes of vertebrates and insects.**

|  | # of sites | # of alleles | k | Fraction of senescing or entrenched alleles |
|---|---|---|---|---|
| **Vertebrates** |  |  |  |  |
| $\omega < 1$ | 2,598,701 (87.8%) | 574,137 (36.4%) | 62.3 (25.1, 100.0) | 0.99 (0.95, 1.00) |
| $\omega = 1$ | 348,047 (11.8%) | 917,340 (58.1%) | 45.3 (−36.8, 98.9) | 0.42 (0.15, 0.90) |
| $\omega > 1$ | 13,189 (0.4%) | 86,852 (5.5%) | −23.3 (−94.7, −3.5) | 0.18 (0.0, 0.86) |
| Total | **2,959,937 (100%)** | **1,578,329 (100%)** |  |  |
| **Insects** |  |  |  |  |
| $\omega < 1$ | 2,699,432 (93.2%) | 429,800 (48.8%) | 47.8 (−1.1, 87.9) | 0.81 (0.47, 1.00) |
| $\omega = 1$ | 185,829 (6.4%) | 413,101 (46.8%) | 9.4 (−80.2, 91.4) | 0.16 (0.01, 0.92) |
| $\omega > 1$ | 9698 (4.4%) | 38,577 (4.4%) | −22.6 (−88.7, 52.0) | 0.25 (0.03, 0.92) |
| Total | **2,894,959 (100%)** | **881,478 (100%)** |  |  |

The analyzed datasets, the corresponding estimates of the rate of entrenchment ($k > 0$) or senescence ($k < 0$) and the fraction of alleles that experience these processes. The values show the median of the ABC posterior distribution of parameter values; numbers in parentheses represent the 95% posterior probability intervals. The total number of sites and alleles in both datasets is shown in bold.

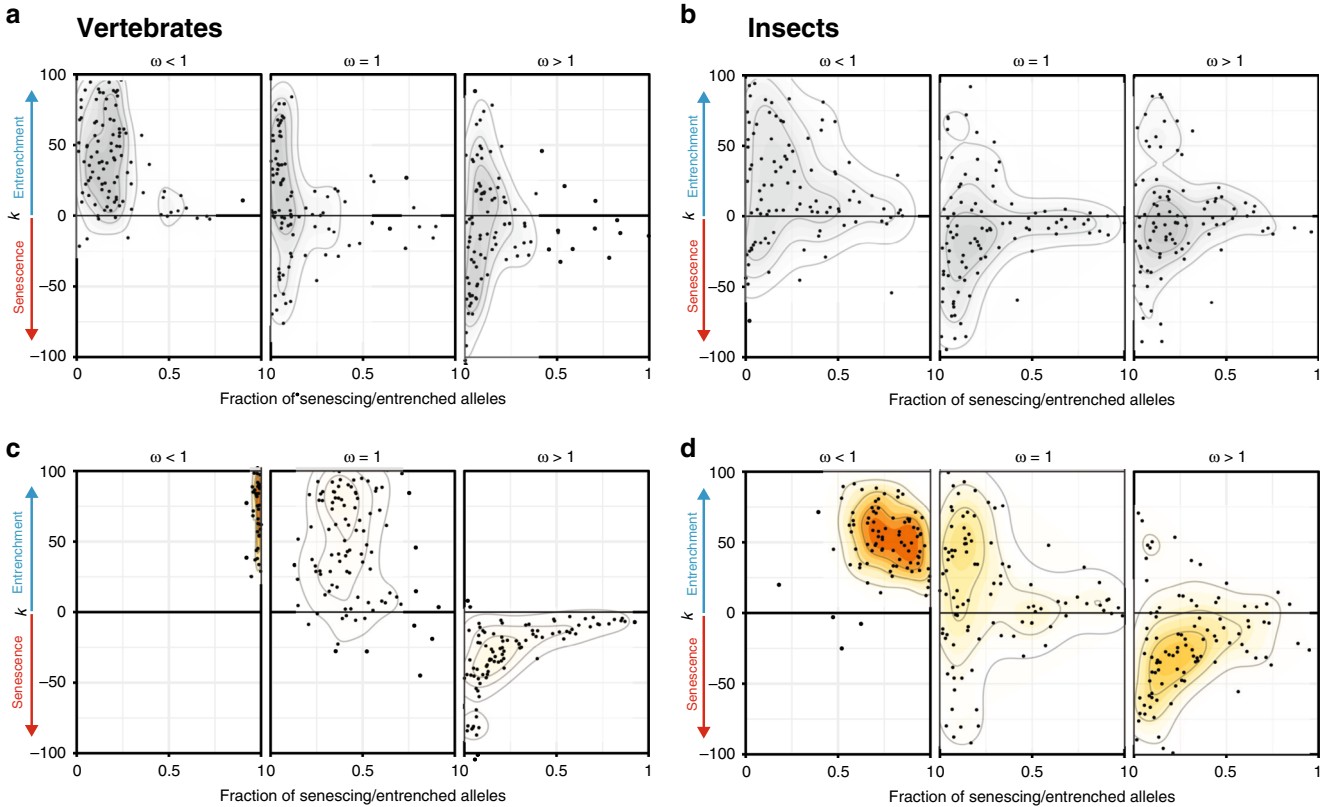

**Fig. 5 Senescence and entrenchment in protein sequences of vertebrates and insects.** Plots provide ABC estimates of the rate of senescence or entrenchment k and the fraction of alleles with changing fitness for protein sequences of vertebrates and insects. For each dataset, the posterior distribution of parameters under 1% acceptance threshold after local ridge regression adjustment is shown. Positive values of k correspond to entrenchment, while negative values of k correspond to senescence of the current allele. **a**, **b** Simulated data under the stationary model using the vertebrate (**a**) or insect (**b**) phylogeny and distribution of ω values. **c**, **d** In real genomic data of vertebrates (**c**) or insects (**d**), sites under negative selection show strong entrenchment, neutral sites demonstrate the intermediate signal, and positively selected sites are senescing.

and selection mode imply that these phenomena are closely related, and that senescence causes many of the adaptive substitutions.

What causes allele senescence? Firstly, it can sometimes result from negative epistasis with an allele arising at an interacting site. While this could result in senescence occasionally, on average epistasis results in entrenchment[25,26]. Secondly, senescence could result from changes in selection pressure external to the genome. We have previously described the acceleration of substitutions over the course of allele lifetime in the evolution of influenza A virus[41].

This pattern has been mainly observed at sites associated with avoidance of the host immune system pressure, and we interpreted it as evidence for negative frequency-dependent selection actively disfavoring the current allele[41]. Here, we show that this type of selection is not a prerequisite for senescence. Instead, senescence is expected whenever selection alters without regard to the identity of the current allele. How much of senescence, and positive selection resulting from it, is due to random changes of the fitness landscape, and how much is due to systematic selection against the current allele, can be a subject of further research.

## Methods

**Data**. We used multiple alignments of exons of vertebrates and insects from the UCSC Genome Browser database together with the corresponding phylogenies[52] (Supplementary Fig. 8). Columns with gaps were excluded. From these alignments, we reconstructed the alleles in the internal nodes of phylogenetic trees with *codeml*[53]. We re-estimated the lengths of individual branches as the average frequency of amino acid substitutions per site on this branch. We classified codon sites as negatively selected ($\omega < 1$), neutral ($\omega = 1$), or positively selected ($\omega > 1$) using the Bayes empirical Bayes method as implemented in the *PAML* package[54]. The size of the datasets and the number of sites and substitution subtrees in each bin are shown in Table 1. The mitochondrial dataset consists of the amino acid alignment of five proteins for several thousand metazoan species[14].

**Simulations**. Simulations of amino acid sequence evolution were performed with the SELVa simulator[43]. SELVa is a forward-time Markov chain simulator that allows the user to model sequence evolution along a predefined phylogenetic tree on static or dynamic SPFLs. The user can specify both the shape of SPFL (i.e., the vector of allele fitnesses for a single position in the genome) and the rule for its change. In this work, we used three types of SPFLs for amino acid sites with 20 possible alleles: flat SPFL corresponding to neutral sites (no substitution leads to change of fitness, log fitness vector is (0, 0, …, 0)); rugged SPFL (one allele is highly preferable over the other ones, log fitness vector is (10, 0, …, 0)) and gamma-distributed SPFL, where the log fitness values for alleles are randomly chosen from the gamma distribution with user-defined parameters.

We used two modes of SPFL change. In the random change mode, the SPFL changes are a Poisson process with a user-defined rate. In this case, fitness values are either reshuffled between alleles (for the flat or rugged SPFL) or redrawn from the same distribution (for gamma-distributed fitnesses). In the current allele-dependent mode, the log fitness of the current allele increases or decreases linearly with time. The user can define the rate of this change ($k$) and the length of the time interval between changes ($\Delta t$). The log fitness of the current allele B at time $t + \Delta t$ is then set to

$$f_B(t + \Delta t) = f_B(t) + k*\Delta t.$$

The fitness values for other alleles remain unchanged. Positive values of $k$ correspond to entrenchment of the current allele, which means its fitness increases with time, and negative ones, to senescence, so that its fitness decreases. When a substitution occurs, i.e., the current allele is replaced with another one, the fitness of the replaced allele stops changing, and the fitness of the new allele starts to change at rate $k$. In this work, we used $\Delta t = 0.01$ dS, where dS is the length of time required for a single substitution at a neutral site; this is small enough to simulate the gradual change of allele fitness.

Individual sites are simulated independently, and interactions between them are not modeled directly. We also do not explicitly model the heterogeneity in substitution rates between alleles, although such heterogeneity arises from the differences in the fitness values of individual alleles drawn from the underlying distribution.

To demonstrate the effects of randomly changing SPFLs, patterns of substitutions emerging due to senescence and entrenchment, and the influence of allele heterogeneity, we used a simple model phylogenetic tree (Supplementary Fig. 12). For ABC-based inference of the dynamics of the current allele fitness in the evolution of real-life protein sequences, we used the corresponding phylogenies of vertebrates and insects (Supplementary Fig. 8).

**Substitution subtrees**. For every replacement A → B on any internal branch of the phylogeny, we can define the corresponding substitution subtree, namely, the contiguous segment of the phylogeny where every internal node carries the derived variant B. For this subtree, B is the current allele, and the initial substitution A → B is the allele gain (Supplementary Fig. 2). Within a substitution subtree, the current allele B can be replaced by the ancestral variant A or some other variant C in the course of allele loss(es).

A genomic position can carry no substitution subtrees if it is fully conservative, or carry one or more substitution subtrees; the number of substitution subtrees equals the number of substitutions on the internal phylogenetic branches in this position. This means that rapidly evolving sites carry more substitution subtrees than conservative ones.

We define a statistic $s_{branch}$, the frequency at which the allele B that has occupied the considered genomic position at the origin of a specific branch has been replaced at this branch. If a single site is analyzed (as in the analysis of mitochondrial genes), $s_{branch}$ can take the values of 0 or 1; if multiple sites are pooled, $s_{branch}$ for different substitution subtrees are considered separately, and $s_{branch}$ can also take values between 0 and 1. $s_{branch}$ is determined by the SPFL and the overall mutation rate of allele B. If the mutation rate is assumed to be constant, changes in $s_{branch}$ with time since the origin of B within the substitution subtree can be used to detect changes in SPFL. We estimate the rate of replacement of B as a function of its age, i.e., the evolutionary time since it was gained.

**Inference of fitness changes for groups of alleles**. An individual substitution subtree usually does not provide enough data to infer SPFL changes. To identify such changes with confidence, we have to pool data across subtrees and sites.

However, pooling data on different subtrees creates a spurious signal of entrenchment due to heterogeneity of evolution rate and unevenness of SPFLs[28,46]. One approach to adjust for these confounding factors is to estimate the mean substitution rate of a subtree, which combines the mutation rate of the site and the effect of the fitness variance of the current allele, and to use this value for model fitting, e.g., in the maximum likelihood (ML) framework. However, our phylogenies are not deep enough to perform ML estimates: the number of substitutions per subtree is too low, while the variance of branch lengths is too large. Instead, to account for confounding effects, we used the approximate Bayesian computation approach (ABC).

Since the age-dependent patterns of substitutions are sensitive to data heterogeneity and the shape of SPFLs, we can't directly measure the rate at which the fitness of the current allele changes. To estimate the strength and abundance of senescence and entrenchment in the evolution of protein sequences, we used rejection ABC (approximate bayesian computation) with ridge regression adjustment as implemented in the *abc* package for R[55]. ABC is a popular method for parameter inference if the likelihood function is not known.

Simulations for ABC were also performed with SELVa. Importantly, rather than simulating the full phylogenetic trees, we simulated individual substitution subtrees. For each dataset of interest, we extracted the list of subtrees generated by substitutions (allele gain events) in this dataset. For each substitution subtree, we then ran SELVa with the given parameters, assuming that the number of sites in the simulation equaled the number of cases when this subtree appeared in the data. The results were pooled across subtrees, and summary statistics were calculated. This approach has two advantages in comparison to simulations based on full phylogenetic trees. First, our summary statistics are based on subtrees only, and using the list of substitution subtrees from the data is more informative than simply the number of sites: this way, we don't have to wait until the ancestral substitution occurs in a simulation, but can start the simulation at the moment we know it has occurred in the data. Second, since the subtrees are smaller, the simulation runs faster.

We used two model functions for ABC. The first one is based on the assumption that all sites in the dataset are susceptible to senescence or entrenchment of the same strength (two-parameter model). It requires two parameters: *alpha* rate parameter for the gamma distribution of alleles fitness values (see above) and the rate of change of the fitness of the current allele $k$.

The second model represents a mixture of two categories of sites: those with a static SPFL ($k = 0$) and those under senescence ($k < 0$) or entrenchment ($k > 0$). It takes three parameters as input: in addition to *alpha* and $k$, it uses the fraction of substitution subtrees (which corresponds to the fraction of alleles) under senescence or entrenchment with rate $k$. The simulated values for $k$ were distributed uniformly from −100 to 100; the fraction of alleles under senescence or entrenchment was also distributed uniformly from 0 to 100%; and *alpha* was distributed log-uniformly from −1.5 to 1.

The number of amino acid sites in the datasets with different site-specific $\omega$ values, and the number of substitution subtrees at these sites, vary between $10^3$ and $10^6$ (Table 1). To account for the variance in summary statistics for smaller datasets, the ABC model function takes a list of subtrees and their counts in the given dataset as input and generates simulations of the same size (but not larger than 100,000 substitution subtrees due to runtime restrictions). For all datasets, we used ridge regression algorithm for parameter estimation as implemented in the *abc* package.

**Summary statistics**. After evaluating a range of possible summary statistics for ABC, we ended up using two statistics based on the dynamics of allele replacement. All branches across all subtrees in the simulation are pooled together and used to calculate the following linear regression:

$$s_{branch} = a*\text{length}_{branch} + b*\text{age}_{branch} + c,$$

where $s_{branch}$ is the frequency at which the current allele was lost lost on the given branch, length$_{branch}$ is the average length of this branch across all substitution subtrees, and age$_{branch}$ is the age of the current allele, i.e., the distance from the root of the substitution subtree to the branch. As summary statistics, we used the values of $a$ and $b$. Supplementary Fig. 11 shows how $a$ and $b$ depend on the simulation parameters: they strongly depend on the direction and the rate of current allele fitness change $k$ and the fraction of alleles susceptible to these changes. Summary statistics were calculated with python scripts using numpy, sklearn, Bio and ete3 packages.

**ABC validation**. Since our method is based on time-dependent dynamics of substitutions, its efficiency is expected to depend on the depth of the phylogenetic tree: short trees will not provide the timescale sufficient to detect changes of the current allele substitution rate. We examined the accuracy of the two-parameter model using test phylogenies with simple topology and internal edges of varying lengths (Supplementary Fig. 3a). Indeed, both the prediction error and the width of the 95% posterior probability interval are the smallest for the phylogenies with internal branches of intermediate lengths (Supplementary Fig. 3c, d). While the shortest trees don't provide the time range sufficient to detect SPFL changes, the largest ones lack resolution and also result in bigger prediction error.

We validated the ABC pipeline for parameter inference using SELVa simulations based on the reconstructed phylogeny of 53 vertebrates and the *abc* package for R[55]. To cross-validate ABC performance under different tolerance rates

and to evaluate the accuracy of parameter estimation for both two-parameter and three-parameter models, we calculated prediction error for parameters based on 100 randomly chosen simulations with the cross-validation function of the *abc* package. The prior size was $10^4$ simulations for both models. The cross-validation results are shown in Supplementary Table 3 and Supplementary Figs. 4 and 5. The tolerance level 0.01 was chosen for both models. Cross-validation tests for parameter inference with our ABC pipeline show that we can accurately estimate the parameters of both models using the selected set of summary statistics.

Next, we asked whether our method is sensitive to changes in the overall rate of evolution. For each model, we generated the testing set of 100 simulations with randomly chosen parameters with normal and twofold increased substitution rate and then used ABC to infer the parameters. We demonstrate that, although the magnitude of $k$ was overestimated for simulations with accelerated evolution rate, the estimates were not biased in any direction ($t$-test $p$ value = 0.53, Supplementary Figs. 6 and 7, top panel; Supplementary Table 2). While the fraction of senescing or entrenched alleles in the three-parameter model was overestimated for simulations with accelerated evolution rate, the magnitude of the bias was not large (on average 0.10, $t$-test $p$ value = 3e−10).

We also checked whether out method allows to confidently distinguish between senescence and entrenchment. The confusion matrices based on the same testing set of simulations show that the frequency of misclassification is 0% for the two-parameter model and 1% for the three-parameter model, and cases of misclassification were only observed in simulations with low $k$ (<1) (Supplementary Table 2). Furthermore, in the few erroneously classified cases, the 95% probability interval for $k$ overlapped with zero.

SELVa stores the sequences of internal nodes of the phylogenetic tree (the ancestral sequences) so that the history of simulated amino acid replacements is known exactly. However, for real data, we use *codeml* to reconstruct the ancestral sequences, and this reconstruction can be erroneous. To make sure that the ancestral state reconstruction does not affect the accuracy of parameter estimation, we reconstructed the ancestral sequences generated by SELVa on the basis of the sequences of terminal nodes in the same way as it was done for the actual data, and used ABC to estimate the parameters using the same procedure as above (Supplementary Figs. 6 and 7, bottom panel). We found that ancestral states reconstruction slightly biased both $k$ (by ~3.8, $t$-test $p$ value = 6e−4) and the fraction of alleles with changing fitness (by ~0.08, $t$-test $p$ value < 2e−16) upwards, but the confusion frequency remained low (0% for the two-parameter model, 0.5% for the three-parameter model), and the only erroneously classified simulation had a low fraction of entrenched alleles (0.09).

We used *evolver* to simulate datasets under different modes of selection to test whether the artifactual signal of senescence or entrenchment can occur in the stationary model of evolution (Fig. 5a, c).

**Reporting summary**. Further information on research design is available in the Nature Research Reporting Summary linked to this article.

## Data availability

The genomic datasets of vertebrates and insects were downloaded from UCSC database https://hgdownload.soe.ucsc.edu (Rosenbloom et al.[52]). The alignments of mitochondrial genes used in this work were published in https://doi.org/10.1093/gbe/evx025 (Klink and Bazykin[14]).

## Code availability

The simulated prior distributions used in the current study, the summary statistics calculated from the genomic datasets of vertebrates and insects and source code for the analysis are available at https://github.com/astolyarova/senescence-ABC.

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

## Acknowledgements

We thank Dmitrii Ivankov and Alexey Kondrashov for useful comments on drafts of this paper.

## Author contributions

G.A.B. conceived the study; A.V.S., E.N., V.V.P., A.V.F., A.D.N., and G.A.B. designed research; A.V.S. and E.N. performed research; V.V.P. developed the formal proof that random changes in SPFL decrease the fitness of the current allele; A.V.S. and A.V.P. analyzed data; and A.V.S., A.V.F., and G.A.B. wrote the paper with comments from all authors.

## Competing interests

The authors declare no competing interests.
