## [Peer Review File · Nature Communications]

Reviewers' comments:

Reviewer #1 (Remarks to the Author):

It is becoming clearer that simple models of protein evolution based on time-invariant independent fitness contributions are inadequate. This raises two questions: What are the various types of models that might be more appropriate? Which models best match observed patterns of evolutionary change? In this paper, Stolyarova and co-workers consider two different models, which they call "senescence" and "entrenchment" and compare the expected results of these models with a phylogenetic analysis of mitochondrial as well as vertebrate and insect genes.

The work is both interesting and potentially important. I have some concerns, which I very much hope the authors can satisfy.

One concern is that there might be a range of models that would result in senescence. Observations of senescence, while interesting, would not necessarily indicate that the fitness is changing in a random manner. It might be useful to consider what other models might result in the same phenomenon, what is necessary for senescence.

In particular, isn't the connection between senescence and positive selection somewhat tautological? Positive selection, whether diversifying or directional, reflect a change in the SPFLs. Changes in the SPFLs result in a declining autocorrelation function. Which is exactly what is manifested by senescence. Or alternatively, sites characterized by an increasing rate of substitutions would be likely identified as under positive selection.

Technically, if I understand correctly, the authors test their ABC method to see whether it can estimate the parameters used to generate simulated data, using the same underlying model for both the data and the estimation. Yet the model is, certainly, not an accurate reflection of the evolution of real proteins. Why should we have confidence that the parameters derived using an erroneous model are interpretable? Is it possible to generate data with a different type of model, and show that the results obtained with the ABC method are robust to this difference?

A few more specific problems

Figure 1: The legend is inadequate. I could not make sense of a, b, and c. What is being shown here? I am guessing that this is a heat map of fitness values, but if so, a scale would be helpful. What is going on in the middle plot of b? Why do substitutions seem to avoid the most favorable amino acid? In d-i, what do the colors of the lines signify?

Figure 2: In the case of a "flat" SPFL, it would seem that no changes in fitness are possible. In this case, senescence and entrenchment do not have meaning. Yet Figure 2 b and c seem to demonstrate that the frequency of substitutions depends on replacing fitnesses with identical values. This does not make sense.

What does "cases when the same allele has originated more than once follow the phylogenetic distribution indicative of it" mean?

Reviewer #3 (Remarks to the Author):

The manuscript entitled "Senescence and entrenchment in evolution of amino acid sites" presented interesting evolutionary phenomenon using bioinformatics characterizations. The manuscript is well written and easy to follow and figures are appropriate too. I really like the manuscript that showed interesting observation of time dependent mutational accumulation in different sites, which could be related to entrenchment and senescence. The work is inspirational for many evolutionay biologists and promote the subsequent experimental studies. I highly recommend the work to publish in Nature communications.

Reviewers' comments:

Reviewer #1 (Remarks to the Author):

It is becoming clearer that simple models of protein evolution based on time-invariant independent fitness contributions are inadequate. This raises two questions: What are the various types of models that might be more appropriate? Which models best match observed patterns of evolutionary change? In this paper, Stolyarova and co-workers consider two different models, which they call “senescence” and “entrenchment” and compare the expected results of these models with a phylogenetic analysis of mitochondrial as well as vertebrate and insect genes.

The work is both interesting and potentially important. I have some concerns, which I very much hope the authors can satisfy.

One concern is that there might be a range of models that would result in senescence. Observations of senescence, while interesting, would not necessarily indicate that the fitness is changing in a random manner. It might be useful to consider what other models might result in the same phenomenon, what is necessary for senescence.

We fully agree with the Reviewer that there are multiple models that would result in senescence. Firstly, senescence at a site A can sometimes result from epistatic interactions if a substitution at an interacting site B disfavors the extant variant in A. While entrenchment, and not senescence, is expected on average (Shah et al. 2015, Pollock et al. 2012), there can certainly be senescing outliers. However, it is unclear why this model should lead to excess of senescence at positively selected sites. This is now discussed.

Therefore, selection pressure external to the genome appears more plausible. While the model with random changes of fitness seems most parsimonious, it is clearly not the only option. We simulate two models: one corresponding to random changes in fitness (Fig. 1), and one corresponding to monotonic reduction in fitness of the current allele (Fig. 2), and show that both models result in senescence. It is difficult to formally distinguish between these models, and we do not attempt this. The former model appears to be simpler and less restrictive; however, the latter model appears more plausible in some biological systems, e.g., under strong frequency-dependent selection against the extant variant, as observed in the influenza A virus (Popova et al. 2019). We now discuss this, and explicitly state that

senescence is consistent with different modes of selection. This does not undermine our main claim that senescence underlies adaptive evolution.

In particular, isn't the connection between senescence and positive selection somewhat tautological? Positive selection, whether diversifying or directional, reflect a change in the SPFLs. Changes in the SPFLs result in a declining autocorrelation function. Which is exactly what is manifested by senescence. Or alternatively, sites characterized by an increasing rate of substitutions would be likely identified as under positive selection.

Thank you for this important comment, which has motivated us to add new material and discussion. We show that the connection between senescence and positive selection is NOT tautological.

First, there can be senescence without positive selection - e.g. if the fitness of the optimal allele drops but it still remains the best one. This is observed in simulations; in the right panel of Fig. 2c, the overall substitution rate at senescing sites is still lower than neutral, indicating that negative selection still prevails.

Second, positive selection without senescence is also conceivable, and there can even be repeated rounds of such selection - i.e., if a constant selection coefficient is associated with each subsequent substitution, as in a "stairway-to-heaven" or similar landscape (Gerrish and Lenski 1998, Desai and Fisher 2007, Kryazhimskiy et al. 2009). Importantly, the existing models of positive selection do not imply senescence, and senescence is not observed in them. To show this, we now add standard markov-chain simulations of positive selection, and show that senescence is not observed in them (Fig 5a,c).

Finally, we now show that the overall substitution rate at senescing and entrenched sites in mitochondrial proteins is similar (Fig. S13), again arguing that rate non-stationarity (senescence or entrenchment) is distinct from the rate itself.

Technically, if I understand correctly, the authors test their ABC method to see whether it can estimate the parameters used to generate simulated data, using the same underlying model for both the data and the estimation. Yet the model is, certainly, not an accurate reflection of the evolution of real proteins. Why should we have confidence that the parameters derived using an erroneous model are interpretable? Is it possible to generate data with a different type of model, and show that the results obtained with the ABC method are robust to this difference?

Indeed, the robustness of the ABC procedure is an important issue. In the previous version, we showed that the results are robust to biased inference of the overall substitution rates and to errors in ancestral sequence reconstruction (Fig. S6-7). We now also show that our signal does not result from stationary models of selection (Fig. 5a,c).

Furthermore, we now test that we do not get spurious signal on non-stationarity if the priors in our models are off. To show this, we obtain priors under one model, and then use these priors to infer parameters of data simulated under a different model. We model two types of error: errors in tree shape, simulating a different tree with different topology and branch lengths (whole-genome tree in the prior, mitochondrial tree in simulated data; Fig. S14-15); and wrong assumptions regarding the distribution of fitness effects (gamma distribution for the prior, lognormal distribution in simulated data; Fig. S16). We don't observe false positive signal of senescence or entrenchment for any of the simulated datasets.

A few more specific problems

Figure 1: The legend is inadequate. I could not make sense of a, b, and c. What is being shown here? I am guessing that this is a heat map of fitness values, but if so, a scale would be helpful. What is going on in the middle plot of b? Why do substitutions seem to avoid the most favorable amino acid? In d-i, what do the colors of the lines signify?

We apologize for the difficulties the Reviewer had with this figure, possibly due to a glitch in pdf. a-c indeed show a heatmap of log fitness values, and a scale is provided; we now moved the scale to the bottom of the figure for better visibility. We also revised the caption for clarity. In all six plots with selection (b-c), positively selected substitutions prevail, but one neutral substitution has also happened (the central plot of b). In d-i, the colors signify the frequency of SPFL changes, as indicated in the legend; again, we moved the legend to the bottom of the figure for better visibility.

Figure 2: In the case of a "flat" SPFL, it would seem that no changes in fitness are possible. In this case, senescence and entrenchment do not have meaning. Yet Figure 2 b and c seem to demonstrate that the frequency of substitutions depends on replacing fitnesses with identical values. This does not make sense.

We agree that the original caption could be confusing, and now rewrote it for clarity. The landscape shapes shown at the top of each panel, e.g. the flat landscape in the left panels of b and c, refer to what we start the simulation with. This is now indicated more explicitly. Over the course of simulation, we modeled monotonic changes of the fitness of the current allele with the rate k : negative values of k correspond to senescence, while positive values correspond to entrenchment. Thus, while the simulated “flat” SPFL starts with equal fitness values of all possible alleles, over the course of evolution, the fitness of the current allele changes directionally with k , making the SPFL not flat anymore. We now clarify this.

What does “cases when the same allele has originated more than once follow the phylogenetic distribution indicative of it” mean?

Thank you for this comment. In this sentence, we meant that there is evidence of entrenchment based on the distribution of homoplasies along the phylogenies. We now revised this paragraph for clarity.

Reviewer #3 (Remarks to the Author):

The manuscript entitled "Senescence and entrenchment in evolution of amino acid sites" presented interesting evolutionary phenomenon using bioinformatics characterizations. The manuscript is well written and easy to follow and figures are appropriate too. I really like the manuscript that showed interesting observation of time dependent mutational accumulation in different sites, which could be related to entrenchment and senescence. The work is inspirational for many evolutionay biologists and promote the subsequent experimental studies. I highly recommend the work to publish in Nature communications.

We thank the Reviewer for high assessment of our work.

REVIEWERS' COMMENTS:

Reviewer #1 (Remarks to the Author):

Really nice paper. I very much like the term "senescence".

Only stylistic comment: "Past substitutions follow the phylogenetic distribution indicative of entrenchment". I do not understand what a phylogenetic distribution means.

Reviewer #1 (Remarks to the Author):

Really nice paper. I very much like the term "senescence".

Only stylistic comment: "Past substitutions follow the phylogenetic distribution indicative of entrenchment". I do not understand what a phylogenetic distribution means.

The thank the reviewer for the high assessment of our work and for the comments which helped us to improve our manuscript. We now revise the mentioned sentence for clarity: "past substitutions are suggestive of entrenchment".